# Interior Sound Field Subjective Evaluation Based on the 3D Distribution of Sound Quality Objective Parameters and Sound Source Localization

**DOI:** 10.3390/ma14020429

**Published:** 2021-01-16

**Authors:** Jiangming Jin, Hao Cheng, Tianwei Xie, Huancai Lu

**Affiliations:** 1College of Mechanical Engineering, Zhejiang University of Technology, Hangzhou 310023, China; jjm@zjut.edu.cn (J.J.); xietianwei_svlab@163.com (T.X.); 2College of Science, Zhejiang University of Technology, Hangzhou 310023, China; chenghao_svlab@163.com

**Keywords:** spherical acoustic holography, sound quality objective parameters, matrix mapping

## Abstract

Controlling low frequency noise in an interior sound field is always a challenge in engineering, because it is hard to accurately localize the sound source. Spherical acoustic holography can reconstruct the 3D distributions of acoustic quantities in the interior sound field, and identify low-frequency sound sources, but the ultimate goal of controlling the interior noise is to improve the sound quality in the interior sound field. It is essential to know the contributions of sound sources to the sound quality objective parameters. This paper presents the mapping methodology from sound pressure to sound quality objective parameters, where sound quality objective parameters are calculated from sound pressure at each specific point. The 3D distributions of the loudness and sharpness are obtained by calculating each point in the entire interior sound field. The reconstruction errors of those quantities varying with reconstruction distance, sound frequency, and intersection angle are analyzed in numerical simulation for one- and two-monopole source sound fields. Verification experiments have been conducted in an anechoic chamber. Simulation and experimental results demonstrate that the sound source localization results based on 3D distributions of sound quality objective parameters are different from those based on sound pressure.

## 1. Introduction

An interior sound field is a compact region with all sound sources located outside. Most of the space inside a car cabin or an airplane fuselage can be defined as an interior sound field. Structure vibration-induced low-frequency noise is one of most unacceptable noises [1] and is hard to dissipate by air or absorb by sound absorption material; therefore, low-frequency noise control in an interior sound field is always a challenging task. There are always people in the interior fields. The ultimate goal of interior sound field noise control is to improve the human auditory feeling [1]. Sound quality objective parameters [2,3,4,5] quantify human subjective auditory perception at a specific position in the sound field. Ergodic measurement for sound quality objective calculation is required to evaluate the entire interior sound field; however, this is impossible in engineering applications.

Near-field acoustic holography (NAH) as proposed by Williams [6], has been developed to reconstruct the entire sound field from the limited measurement points in a microphone array. In order to satisfy the conditions of the homogenous Helmholtz equation, the sound field should be a source-free field which means that there should not be any sources in the calculation region. The microphone array is located in the near field of the sound source so as to catch the evanescent waves which radiate by the structure and decay exponentially from the structure surface. Structure-borne sound related to structural vibration can be reconstructed by NAH. Theoretically, the spatial resolution of the reconstructed sound field is unlimited, but in practice the highest-wavenumber sound will be limited by the number of microphones, which determines the distance between the microphones.

Spherical Near-field Acoustic Holography (SNAH) [6], together with a spherical microphone array [7,8,9,10] can measure waves in any direction, which gives this methodology unique advantages in interior sound field reconstruction when internal sound waves propagate in all directions. In 2006 [11] and 2010 [12], Williams et al. proposed a spherical near-field acoustic holography-based sound intensity reconstruction methodology for open and rigid spherical microphone arrays, and applied that methodology to airplane fuselage and car cabin interior sound field evaluation. In 2011, Jacobson et al. [13] proposed a similar method for a rigid spherical microphone array. In 2016, Li et al. [14] developed a method to reconstruct more complex interior sound fields inside a spherical shell.

However, the sound field reconstruction results from SNAH do not reveal the contributions of sound sources to the sound quality objective parameters, suppressing the high sound pressure level source maybe not be able to improve the auditory feeling. To address this issue, Song et al. [15,16,17] used beamforming and the Zwicker [18] loudness model to reconstruct 2D distributions of loudness. Experimental results showed that the highest-sound pressure source and the loudest-sound source were not the same. The output of beamforming, or spherical harmonic beamforming (SHB) [19,20,21], is the directivity of the sound field in the center of the microphone array. In order to determine the sound pressure distribution of the sound field, one scale factor is applied to the output of the beamforming to obtain the sound pressure contributions to the sound field, which can be used to calculate the sound quality objective parameters. Beamforming is appropriate for high frequency (>1000 Hz) sound field analysis. To the best knowledge of the authors, no correlative work has been conducted to reconstruct the 3D distributions of sound quality objective parameters in the sound field when the frequency is below 1000 Hz. Other researchers, such as Gao et al. [22], calculated the 3D spatial distribution of loudness in the low-frequency range using Finite Element Analysis (FEA) vibro-acoustic simulations, but they have not reconstructed the sound field.

In this paper, a mapping method from the sound-pressure field to sound quality objective parameter 3D distribution fields is presented. Sound sources which are related with human auditory feeling can be identified and localized by this methodology. The theoretical framework of SNAH is explained. The mapping method which is based on the Moore loudness model and the Aures [23] sharpness model is illustrated. The normalized least-square errors for sound field reconstruction varying with the reconstruction distance and frequency range for single monopole source sound field, and the intersection angle between two monopole source fields are analyzed. The mapping method has been verified by experiments with a 36-channel spherical array and one or two monopole sources, which were carried out in an anechoic chamber. The methodology proposed in this paper will be meaningful to characterize the sound absorption [24] or insulation [25] properties of foam polymers, which were laid on boundaries of the interior sound field.

## 2. Methods

### 2.1. Theoretical Model for Spherical Acoustic Holography with a Rigid Spherical Array

The theory of spherical acoustic holography [11,13] is illustrated in this section. In the source-free sound field shown in Figure 1, when one rigid spherical object is introduced into the interior sound field, the incident waves generated by the sources will be scattered by the rigid surface of the spherical microphone array, and the total sound pressure *p_t_* is the superposition of the incident sound wave *p_i_* and the scattered wave *p_s_*.
(1)$$p_{t}=p_{i}+p_{s}$$

According to the Helmholtz equation in spherical coordinates, the solution for the incident wave can be written as
(2)$$p_{i}(r,\theta ,\varphi )=\sum_{n=0}^{\infty }\sum_{m=-n}^{n}A_{mn}j_{n}(kr)Y_{n}^{m}(\theta ,\varphi )$$
where (*r*, θ, φ) is a specific field point; *k* is the wave number, k=ω/c; *c* is the sound speed; $j_{n}(kr)$ is the spherical Bessel function; and $Y_{n}^{m}(\theta ,\varphi )$ is the spherical harmonic. $A_{mn}$ is the incident wave coefficient of the spherical wave. The solution for scattering waves is
(3)$$p_{s}(r,\theta ,\varphi )=\sum_{n=0}^{\infty }\sum_{m=-n}^{n}C_{mn}h_{n}(kr)Y_{n}^{m}(\theta ,\varphi ),(r\geq a)$$
where $C_{mn}$ is the scattering coefficient of the spherical wave and $h_{n}(kr)$ is the Hankel function of the first kind.

If a Neumann boundary condition for the rigid spherical array with radius *a* is applied, the particle velocity on the rigid surface of the spherical array should be zero.
(4)$$w(a)\equiv \frac{\partial p_{t}}{\partial r}|{}_{r=a}\equiv \frac{\partial \left(p_{i}+p_{s}\right)}{\partial r}|{}_{r=a}=0$$
where w(a) is normal partial velocity, and the scattering coefficient should be
(5)$$C_{mn}=-A_{mn}\frac{j_{n}^{\prime }(ka)}{h_{n}^{\prime }(ka)}$$

The superscript’ denotes the derivation. The scattering sound pressure in the sound field can be expressed [13]:(6)$$p_{s}(r,\theta ,\varphi )=-\sum_{n=0}^{\infty }\frac{j_{n}^{\prime }(ka)}{h_{n}^{\prime }(ka)}h_{n}(kr)\sum_{m=-n}^{n}A_{mn}Y_{n}^{m}(\theta ,\varphi ),(r\geq a)$$

Therefore, the total sound pressure *p_t_* can be written as
(7)$$p_{t}(r,\theta ,\varphi )=\sum_{n=0}^{\infty }\left(j_{n}(kr)-\frac{j_{n}^{\prime }(ka)}{h_{n}^{\prime }(ka)}h_{n}(kr)\right)\sum_{m=-n}^{n}A_{mn}Y_{n}^{m}(\theta ,\varphi ),(r\geq a)$$

In order to calculate the unknown coefficient $A_{mn}^{}$, we multiply by the complex conjunct $Y_{n}^{m}{\left(\theta ,\varphi \right)}^{*}$ on both sides of Equation (7), and integrate the equation on the spherical surface *r* = *a.* The spherical harmonic functions are orthogonal; therefore, only the $Y_{n}^{m}(\theta ,\varphi )$ term’s coefficient $A_{mn}^{}$ will be left after the integration. For numerical calculation, the continuous spherical integration must be replaced by a discrete spherical surface sum, and the approximate coefficient ${\tilde{A}}_{mn}$ can be written as
(8)$${\tilde{A}}_{mn}=\frac{\sum_{i=1}^{M}w_{i}p_{t}(a,\theta_{i},\varphi_{i})Y_{n}^{m}{\left(\theta_{i},\varphi_{i}\right)}^{*}}{j_{n}(kr)-\frac{j_{n}^{\prime }(ka)}{h_{n}^{\prime }(ka)}h(kr)}$$
where $p_{t}(a,\theta_{i},\varphi_{i})$ is the scattering sound wave at the *i*th position, $w_{i}$ is the spherical integral coefficient, and *M* is the number of microphones.

Finally, the incident sound pressure *p_i_* can be rewritten as
(9)$$p_{i}(r,\theta ,\varphi )=\sum_{n=0}^{N}\sum_{m=-n}^{n}{\tilde{A}}_{m\;n}j_{n}(kr)Y_{n}^{m}(\theta ,\varphi )$$
where *N* is the number of spherical harmonic terms, which depends on the number of microphones *M* and the sound frequency fRec. A more accurately reconstructed sound field be obtained if more microphones are used. As shown in the error analysis section, with 36 microphones in a rigid spherical array, *N* = 5 gives the best sound field reconstruction accuracy, and this truncation number is used in all calculation results throughout this paper. The most complex case in this paper is the two-monopole sound field; therefore, the series-truncation regularization method was applied in this study. If we need to deal with a much more complex sound field, advanced regularization methods, such as Tikhonov regularization, should be employed.

### 2.2. Mapping from Sound Pressure to Sound Quality

Mapping the sound pressure field to the objective parameter field of the sound quality gives not only the three-dimensional distribution of the conventional acoustic quantity (sound pressure value), but also the three-dimensional distribution of the sound quality objective parameters. It reveals the relationship between the location of the noise source and the subjective auditory feeling of listeners.

According to the loudness definition in ANSI3.4-2007 [3], all audible sound is included in the loudness model, but for a specific spherical microphone array, only the sound in the effective frequency band can be reconstructed, and the out-of-band sound will not be included. The spherical microphone array is a special design for one particular sound field where the major frequency components lie inside the effective frequency band of the microphone array. For simplicity, the free field assumption is also adopted.

The procedure for calculating 3D distributions for sound quality objective parameters is:Calculate the sound pressure *P_t_* in the frequency domain for each point by FFT, in the validated frequency band of the spherical microphone array;Calculate the incident sound pressure *P_i_* by SNAH;Calculate the loudness and sharpness by using a mapping matrix for a single point;Obtain the 3D distributions of sound quality objective parameters through computing the sound quality objective parameter value at each calculation point in space.

The loudness mapping matrix for a specific point in the sound field can be written as
(10)$${\left[p_{i}(r,\;\theta ,\;\varphi ,\;\omega_{1})\cdots p_{i}(r,\;\theta ,\;\varphi ,\;\omega_{m})\right]}_{1\times m}{\left[\begin{array}{c} W_{1}\cdots W_{372}\\ W_{1}\cdots W_{372}\\ \cdots \\ W_{1}\cdots W_{372} \end{array}\right]}_{m\times 372}={\left[N_{1}^{\prime }(r,\;\theta ,\;\varphi )\cdots N_{372}^{\prime }(r,\;\theta ,\;\varphi )\right]}_{{}_{1\times 372}}$$

The matrix form is
(11)$$PW=N^{\prime }$$
where *P* is the vector of calculated sound pressures at different frequencies, and *W* is a matrix of 372 specific loudness models, each one corresponding to each critical band, and the audible frequency range is divided into those 372 critical bands. *N*′ is the specific loudness vector. Loudness can be calculated by summing for all specific loudness values *N*′.

The Aures sharpness of each point in sound field is calculated using the Zwicker loudness, where only 24 critical bands are used in the Zwicker loudness model. The 3D distribution can be obtained by calculating the sharpness value at each point.
(12)$${\left[p_{i}(r,\;\theta ,\;\varphi ,\;\omega_{1})\cdots p_{i}(r,\;\theta ,\;\varphi ,\;\omega_{m})\right]}_{1\times m}{\left[\begin{array}{c} u_{1}\cdots u_{24}\\ u_{1}\cdots u_{24}\\ \cdots \\ u_{1}\cdots u_{24} \end{array}\right]}_{m\times 24}={\left[N_{1}^{\prime }(r,\;\theta ,\;\varphi )\cdots N_{24}^{\prime }(r,\;\theta ,\;\varphi )\right]}_{{}_{1\times 24}}$$

The matrix form is
(13)$$PU=N^{\prime }$$
where *U* is the 24 critical frequency band filters and *N*′ is the Zwicker specific loudness vector. The Zwicker loudness is the summation of the Zwicker specific loudness vector, and the Aures sharpness can be calculated from the summation of the Zwicker specific loudness vector. 

The sharpness model based on the Zwicker loudness model is
(14)$$S=0.11\times \frac{\int_{0}^{24Bark}N^{\prime }g(z)zdz}{N}$$
where the specific loudness *N*′(*z*) for the 24 critical frequency bands can be calculated from the Zwicker loudness model. The coefficient *g*(*z*) for each critical frequency band can be determined from
(15)$$g(z)=\left\{ \begin{array}{ll} 1, & z\leq 16\\ 0.06e^{0.17z}, & z>16 \end{array}\right.$$

### 2.3. Analytical Solution

Once the monopole sound source is located at point $\left(r_{s},\theta_{s},\phi_{s}\right)$, the sound pressure at each microphone position (r,θ,φ) of the spherical array can be calculated from
(16)$$p_{t}\left(r,\theta ,\varphi \right)=-ik^{2}\rho cQ_{s}\sum_{n=0}^{N_{s}}\left[j_{n}\left(kr_{<}\right)h_{n}\left(kr_{>}\right)-\frac{j_{n}\left(kr_{<}\right)}{h_{n}\left(kr_{>}\right)}h_{n}\left(kr\right)h_{n}\left(kr_{s}\right)\right]\sum_{m=-n}^{n}Y_{n}^{m}\left(\theta ,\varphi \right)Y_{n}^{m}\left(\theta ,\varphi \right)*$$
where *ρ* is the air density and *N*_s_ is the truncation number of the spherical harmonic; in this paper, *N*_s_ = 30. The notation *r*_<_ represents the smaller of *r* and *r*_s_, and *r*_>_ represents the larger of *r* and *r*_s_. When these values are used as input data, the sound field in the space surrounding the array can be reconstructed by Equation (9).

### 2.4. Normalized Least-Square Errors

If the number of calculation points for total sound fields is *J*, the normalized least-square error for sound pressure reconstruction *Error.P* is
(17)$$Error.P(\%)=\frac{\sum_{j=1}^{J}{\left({\left\|p_{i}(j)-p(j)\right\|}_{2}\right)}^{2}}{\sum_{j=1}^{J}{\left({\left\|p(j)\right\|}_{2}\right)}^{2}}\times 100\%$$
where $p_{i}(j)$ is the reconstructed sound pressure of the incident wave at point *j*, and p(j) is the theoretical sound pressure of the incident wave at point *j*.

The definition for the normal least square errors for loudness reconstruction *Error.N* and sharpness reconstruction *Error.S* are similar to that for sound pressure, and are
(18)$$Error.N(\%)=\frac{\sum_{j=1}^{J}{\left({\left\|N_{rec}(j)-N(j)\right\|}_{2}\right)}^{2}}{\sum_{j=1}^{J}{\left({\left\|N(j)\right\|}_{2}\right)}^{2}}\times 100\%$$
(19)$$Error.S(\%)=\frac{\sum_{j=1}^{J}{\left({\left\|S_{rec}(j)-S(j)\right\|}_{2}\right)}^{2}}{\sum_{j=1}^{J}{\left({\left\|S(j)\right\|}_{2}\right)}^{2}}\times 100\%$$
where *N_rec_*(*j*), *S_rec_*(*j*) are the reconstructed loudness and sharpness calculated from sound pressure at point *j*, and *N*(*j*), *S*(*j*) are the theoretical loudness and sharpness, respectively.

## 3. Simulation

### 3.1. Errors of SNAH

The simplest one-monopole sound field is used as an example, when the source frequency *f* varies between 100–2500 Hz, and air density is 1.3 kg/m^3^, sound speed is 340 m/s, and sound strength *Q_s_* is 3.6 × 10^−5^ Pa·m. The analytical solution for sound pressure in entire interior sound field can be obtained from Equation (16).

With the sound pressure in microphone positions of the rigid spherical array used as the inputs, the SNAH sound pressure *p_i_* on the reconstruction spherical surface is reconstructed by SNAH. The sound pressure calculated by Equation (9) on the reconstruction spherical surface is used in the analytical solution, and then the reconstruction error for the entire sound field is calculated from Equation (17). 

The sound pressure reconstruction error on the surface *r* = 0.1 m varies with the truncation number and sound frequency, as presented Figure 2. For a given microphone number, the reconstruction error declines at first, and then increases dramatically, and there exists an optimal truncation number *N_opt_*. These phenomena can be explained as follows. If the truncation number is too small, the superposition of the corresponding spherical harmonic function is not accurate enough to represent the complexity of sound field, leading to a large reconstruction error; and if the truncation number is too large, the number of spherical harmonic coefficients ${\tilde{A}}_{n}^{m}$((*N*+1)^2^) is larger than the microphone number *M*, the number of unknowns is greater than number of equations in Equation (2), and the reconstruction errors will increase uncontrollably. The reconstruction error for the high-frequency sound field is greater than that for the low-frequency field, because the wavelength of high-frequency sound is shorter, which means fewer points in one wavelength are measured with a fixed configuration of the array. For the 36 channel spherical microphone arrays used in this paper, *N_opt_* = 5 led to the lowest reconstruction errors for *f* < 1800 Hz.

As shown in Figure 3, with *a* = 0.1 m, *d* = 0.7 m, and *N_opt_* = 5, the reconstruction error grows consistently with the reconstruction distance and sound frequency. When *r* < 0.2 m, *f* < 1200 Hz; or *r* < 0.3 m, and *f* < 800 Hz, the sound pressure reconstruction errors was less than 10%.

### 3.2. Errors for 3D Distribution of Sound Quality Objective Parameters

The 3D distributions of sound quality objective parameters are calculated based on the sound pressure reconstruction; therefore, the sound pressure reconstruction errors will determine the reconstruction accuracy of the 3D distributions of sound quality objective parameters. Figure 4 presents the reconstruction errors for the 3D distributions of sound quality objective parameters varying with reconstruction distance and sound frequency. The SNAH truncation number for spherical harmonics is *N* = 5. The reconstruction errors of the 3D distributions of sound quality objective parameters varying with reconstruction distance are also described in Figure 4. The reconstruction errors of loudness and sharpness both increase together with reconstruction distance. 

The reconstruction error was less than 10% when the reconstruction distance was less than 0.2 m, and the frequency was lower than 1200 Hz. Normalized least-squared errors for the reconstruction 3D distribution of loudness is smaller than sound pressure; the accuracy of loudness reconstruction is obviously improved for low frequency and short reconstruction distance.

For the analysis frequency range 100–1200 Hz, the reconstruction errors for sharpness are much lower than those for sound pressure, because the sharpness is more relevant to high frequency sound (*f* > 1000 Hz) and is related modulated frequency tone, which represents the temporal characteristics of sound. Both quantities are insensitive to variations of sound pressure.

### 3.3. Simulation Results for 3D Distributions of Loudness and Sharpness

In order to evaluate the reconstruction errors for 3D distributions of sound quality objective parameters varying with reconstruction distance and sound frequency, the one monopole sound field with rigid spherical microphone array was reconstructed by the proposed method. The spherical radius *a* was 0.1 m, and the monopole source was located at (0.2 m, pi/2, pi/2). Figure 5 shows the simulation results of the spatial distributions of sound quality objective parameters at source frequency *f* = 100 Hz, at the reconstruction distances of 0.1 m, 0.15 m, and 0.2 m, with a truncation number for spherical harmonics of 5. The location of the sound source can be localized and identified by spatial reconstruction of the sound pressure and sound quality objective parameters. The sound pressure distribution trends and source localization results were unaltered with increasing of the reconstruction distance. The higher values of sound pressure and sound quality objective parameters became larger with the decrease in standoff distance from the sound source.

The influence of sound source frequency on the spatial distribution of sound quality objective parameters is investigated in Figure 6, with spherical radius *a* = 0.1 m, monopole source location (0.2, pi/2, pi/2), and the reconstruction distance 0.1 m. The results are similar to those in Figure 5, and the source localization results remain unaltered with increases in the reconstruction distance.

### 3.4. Two-Monopole Sound Field

In order to further investigate the spatial distribution of the sound quality objective parameter field in the case of a more complicated sound field, it was necessary to carry out the simulation analysis of a stationary two-monopole sound field (Figure 7). The center of the two monopole sound sources and the geometric center of the spherical microphone array are not at the same point, therefore according to Equation (6), the SNAH model with the coordinate origin in the center of microphone array needs the expansion in high-order spherical harmonic functions to approach the actual sound field. Using SNAH to reconstruct the field of two-monopole sources is a complex problem, and it is of general significance to study the 3D distribution of sound quality objective parameters in this sound field.

Figure 7 shows the two-monopole sound field, with spherical array *a* = 0.1 m and reconstruction distance *r* = 0.1 m. The standoff distance from the spherical array to the two-monopoles is *d*_1_ = *d*_2_ = 0.2 m. θ is the intersection angle between the connection lines from the origin to Source 1 and Source 2. Source 1 and Source 2 have the same intensity, but Source 2 has a higher sound frequency than Source 1. The sound pressure on the surface of the spherical array produced from Source 1 is higher than that from Source 2, because high-frequency sound decreases more rapidly than low-frequency sound.

Keeping the other system parameters fixed, we evaluated the 3D distribution of sound quality objective parameters varying with the included angle θ. The calculation results are shown in Figure 8. With the small intersection angle 60°, the two monopole sources were merged together into one source and could not be localized and identified separately. When the intersection angle increased to 90°, those two monopole sources could be localized according to the sound pressure, loudness and sharpness. Specifically, the 3D distributions of loudness and sharpness could much more clearly localize the sound sources. However, the sound source localization results are different, and the second source which had higher frequency and lower sound pressure could also be clearly identified. The intersection angle of 180° produced the similar sound source localization results.

## 4. Experiment 

Due to the directivity of the speaker and measureing errors in the experiment, the proposed methodology needed to be verified by experiment. One- and two- speaker experiments were conducted in the anechoic chamber at the Zhejiang University of Technology (ZJUT). The free-field size of the anechoic chamber was 1.5 m × 1.2 m × 1.5 m, with background noise less than 18 dB, and the cut-off frequency was 63 Hz.

The sound pressure data were measured by a Bruel&Kjaer 4958 microphone (Bruel & Kjaer, Nærum, Denmark), which has 10 Hz–20 kHz frequency range and dynamic range 28–140 dB. A Bruel&KjaerWA1565 rigid spherical microphone array with applicable sound frequency up to 8 kHz was adopted in the anechoic chamber experiment. The distribution of microphones on the surface of the WA1565 rigid spherical microphone array was optimized using a spherical integration method. The positions of the source and array in the experiment are shown in Figure 9. The origin in the mathematical model was set at the center of the spherical microphone array.

The input signal to the speaker was a pure single frequency-harmonic at the same level but different frequencies, which were 1000 Hz and 1200 Hz, because 1200 Hz is almost the highest allowable frequency for the reconstruction distance *r* = 0.1 m according to the error analysis in the simulation section. The time recording span was 10 s, and the sample rate was 51.2 kHz. The reconstruction results for the 3D distributions of sound pressure and sound quality objective parameters for the one-speaker sound field at 1000 Hz and 1200 Hz are shown in Figure 10. From the spatial distribution results of sound pressure and sound quality objective parameters, the sound source could be identified in this experiment.

The excitation frequencies for Speaker 1 and Speaker 2 were 1000 Hz and 1200 Hz, respectively, and the speakers had the same excitation level. The standoff distance from the speaker to the spherical array was 0.2 m. The reconstruction results for the 3D distributions of sound pressure, loudness and sharpness varying with intersection angle are shown in Figure 11. We obtained opposite sound source localization results based on 3D distributions of loudness and sharpness. On the surface of the spherical microphone array, high sound pressure Source 1 produced higher loudness and lower sharpness, while high-frequency Source 2 produced higher sharpness and lower loudness. The methodology of identifying and localizing the sound sources according to the 3D distribution of the sound quality objective parameters can find more annoyance sources, which presents direct evidence for experts to improve the comfort of an interior sound field.

## 5. Conclusions

In this paper, combining the theory of spherical near-field acoustical holography (SNAH) and the calculation model of sound quality (SQ) objective parameters, a joint analysis methodology is proposed. The joint method uses the calculation model of mapping the distribution of the sound pressure field to the sound quality objective parameter 3D distribution. This sound quality objective parameter 3D distribution is used to localize the noise source contributing most to the human subjective auditory perception. The models used to calculate sound quality objective parameters are introduced, and the effectiveness of the algorithm was proved by numerical simulations and experiments in an anechoic chamber. The major factors influencing the sound quality objective parameters are discussed by numerical simulation analysis. The proposed method was validated by simulation and experiment in the anechoic chamber by one- and two- monopole acoustic sources. The main conclusions are: (a) The calculation of objective parameter fields of sound quality are based on the calculation of the sound pressure field. The reconstruction method of sound fields based on the theory of spherical acoustic holography are applicable to the identification and location of the low-frequency sound source. The reconstruction error is less than 10% when the source frequency is below 1200 Hz and the reconstruction radius is less than 0.2 m. (b) When the frequency of the sound source is in the range of 100–1600 Hz, the maximum reconstruction error in the sound pressure is 11%, while the reconstruction error of the objective parameters of sound quality is less than 5%. The variation of the reconstruction errors of objective parameters of sound quality is consistent with that of the reconstruction error of sound pressure, which increases with the reconstruction radius. The proposed method should be further developed in several aspects, such as sound quality subjective analysis and reliability evaluation in more complex acoustic environments—both of which are being studied further in the laboratory. This method is promising for applications such as automatic interior sound source localization and evaluation, characterization of sound absorption [24], or insulation [25] properties of noise control acoustic material [24,25]. 

## Figures and Tables

**Figure 1 materials-14-00429-f001:**
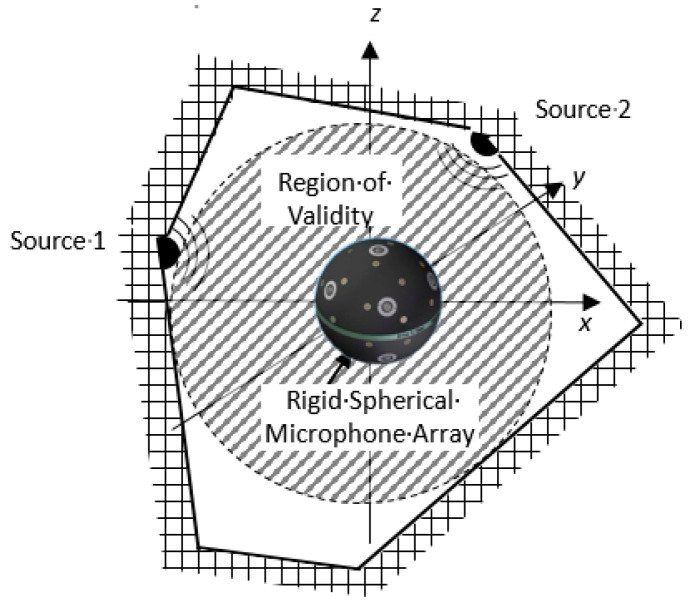
Validity region of spherical acoustic holography.

**Figure 2 materials-14-00429-f002:**
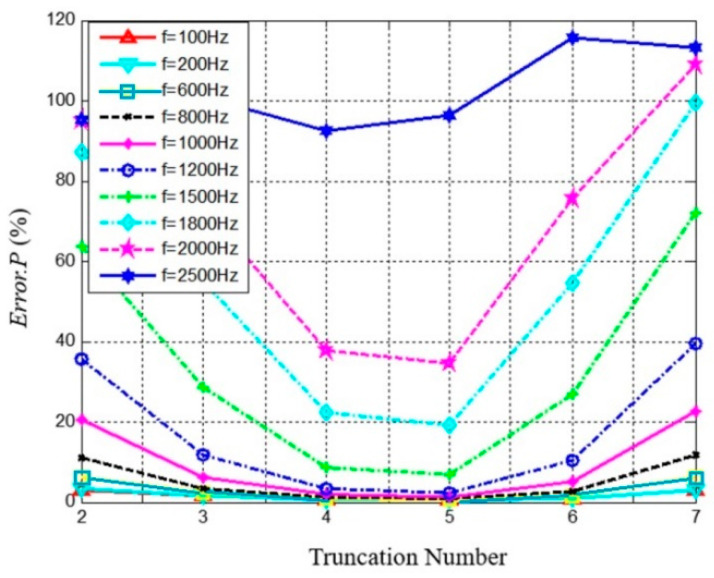
The reconstruction errors of different truncation numbers in different sound frequency (*r* = 0.1 m).

**Figure 3 materials-14-00429-f003:**
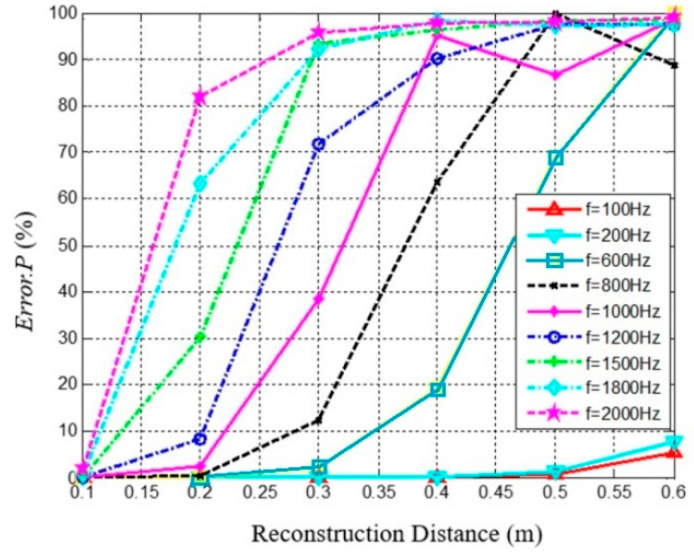
The reconstruction errors vary with reconstruction distance at different sound frequencies (spherical radius *r* = 0.1 m).

**Figure 4 materials-14-00429-f004:**
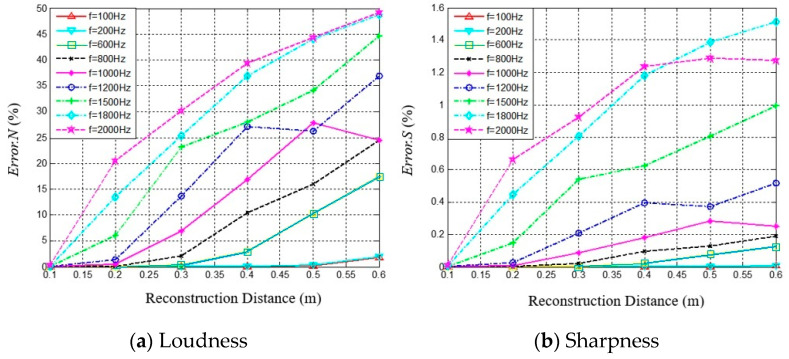
The variation of reconstruction errors of sound quality objective parameters with reconstruction distance at different sound frequencies. (**a**) Reconstruction errors of Loudness, (**b**) Reconstruction errors of Sharpness.

**Figure 5 materials-14-00429-f005:**
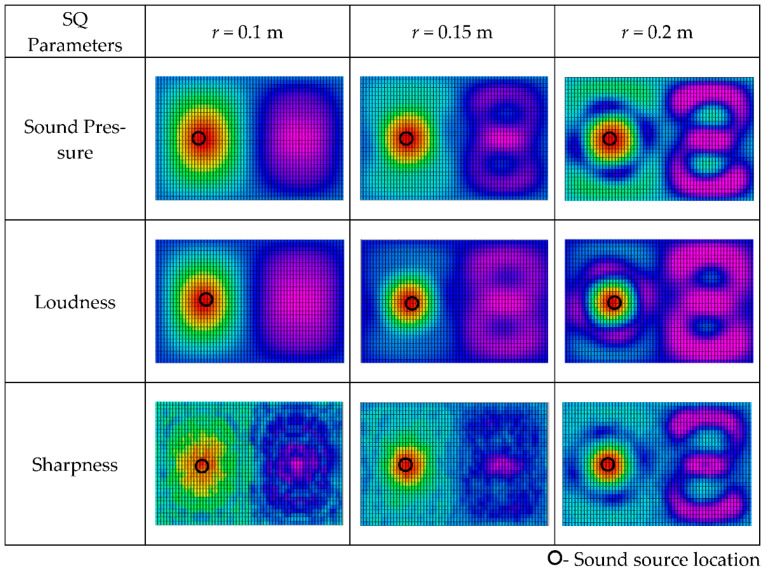
Spatial distribution of sound quality (SQ) objective parameters for source frequency *f* = 100 Hz at different reconstruction distances. (The results are shown in the form of plane expansion of the spherical calculation surface, and the horizontal axis is longitude: −π~π, the vertical axis is latitude: −π/2~π/2).

**Figure 6 materials-14-00429-f006:**
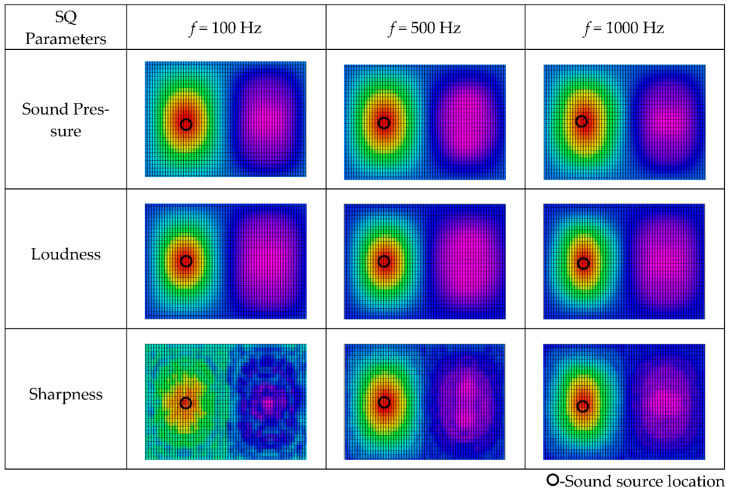
Spatial distribution of sound quality objective parameters for source frequency reconstruction distance *r* = 0.1 m at different sound frequencies. (The results are shown in the form of plane expansion of the spherical calculation surface, and the horizontal axis is longitude: −π~π, the vertical axis is latitude: −π/2~π/2).

**Figure 7 materials-14-00429-f007:**
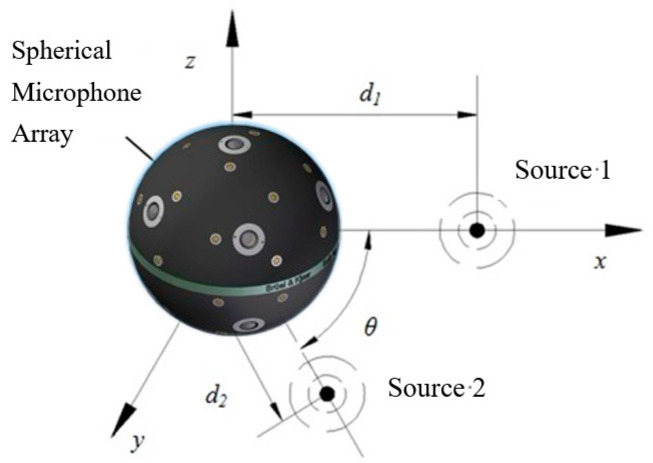
Diagram of two monopole sound field. Spherical microphone array is located in the origin. Two monopole sources are located in the *x*–*y* plane: one is on the *x*-axis, and the second monopole source is located on the first quadrant which has the same standoff distance from the spherical array with the first one—intersection angle between the two source is θ.

**Figure 8 materials-14-00429-f008:**
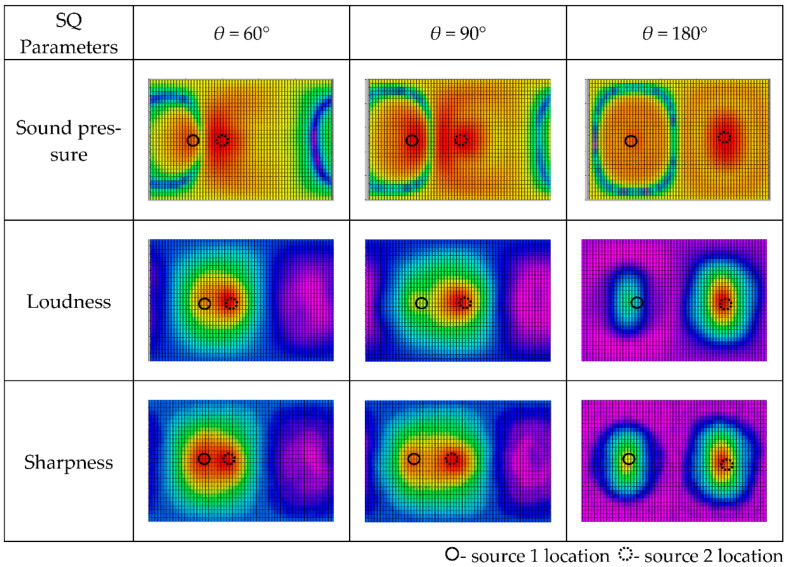
Spatial distributions of sound quality objective parameters for source reconstruction distance *r* = 0.1 m at different sound frequencies. (The results are shown in the form of plane expansion of the spherical calculation surface, and the horizontal axis is longitude: −π~π, the vertical axis is latitude: −π/2~π/2).

**Figure 9 materials-14-00429-f009:**
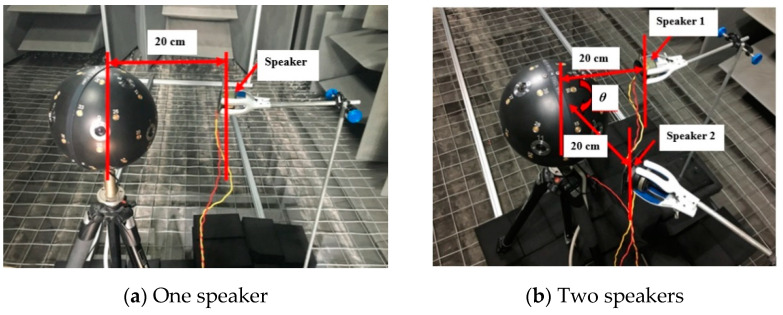
Photographs of the experimental setup.

**Figure 10 materials-14-00429-f010:**
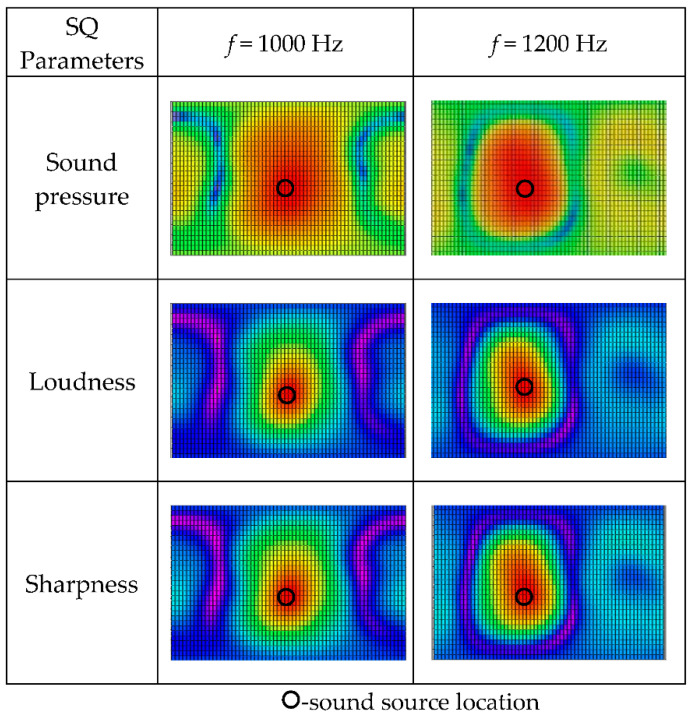
Spatial distributions of sound quality objective parameters for source reconstruction distance *r* = 0.1 m at different sound frequencies. (The results are shown in the form of plane expansion of the spherical calculation surface, and the horizontal axis is longitude: −π~π, the vertical axis is latitude: −π/2~π/2).

**Figure 11 materials-14-00429-f011:**
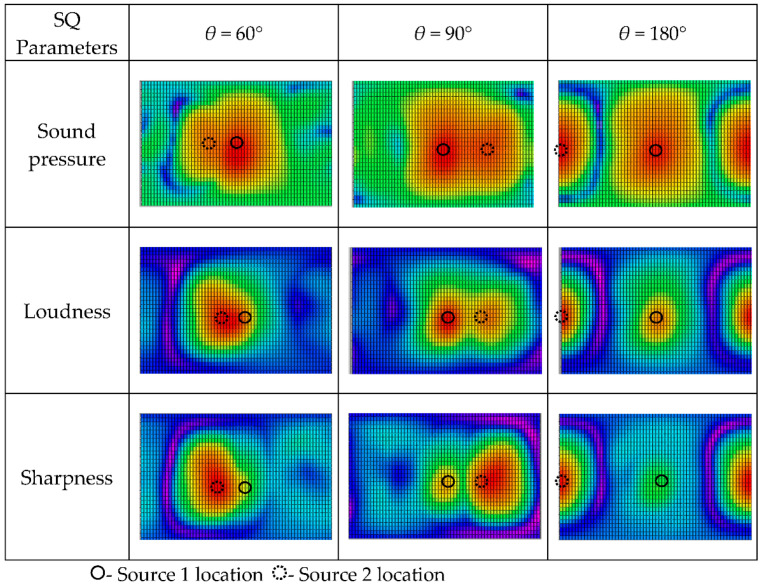
Spatial distributions of sound quality objective parameters varying with intersection angle for source frequencies *f*_1_ = 1000 Hz and *f*_2_ = 1200 Hz, and reconstruction distance *r* = 0.1 m. (The results are shown in the form of plane expansion of the spherical calculation surface, and the horizontal axis is longitude: −π~π, the vertical axis is latitude: −π/2~π/2).

## Data Availability

This article does not contain any additional data.

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
