# Peer review of "Interior Sound Field Subjective Evaluation Based on the 3D Distribution of Sound Quality Objective Parameters and Sound Source Localization"

_materials, 2021, doi:10.3390/ma14020429_

Round 1

Reviewer 1 Report

The paper addresses a topic of technical and scientific interest for researchers in acoustics. Overall, the article is well structured and complete. A linguistic revision is recommended.

Specific comments are given in the table below:

P. Line Comment
2 3 Please clarify the meaning of "evanescent sound field"
2 3 Structure-bone sound -> Structure-borne sound
2 3rd paragraph Please check this phrase: "...suppressing the high sound pressure level source maybe not be able to improve the auditory feeling."
4 After eqn. 2.9 Please check this phrase: “More accurately reconstructed sound field scan be obtained if more microphones are used”
5 2.3 Analytical... Please check this phrase: “For a the monopole sound source is located at point…”
6 Last paragraph is not accurately enough -> is not accurate enough
7 3.2 Errors reconstruction distant -> reconstruction distance
8 1st paragraph Please check: “…is related to modulated frequency tone which is represent the temporal characters of sound…
8 1st paragraph to with variations -> to variations

Author Response

Dear reviewers:

We express our sincere appreciation for your careful reading and helpful comments. Those comments are of great significance for revising and improving our article. We have addressed the points according to your advice and have given the answers to the comments below.

Responses to Reviewer 1:

Point 1: Page 2 Line 3: Please clarify the meaning of "evanescent sound field".

Response 1: Very thanks for your professional concern. The revise for paper is: evanescent wave which radiate by the structure and decay exponentially from the structure surface.

Point 2: Page 2 Line 3: "Structure-bone sound" should be revise as "Structure-borne sound"

Response 2: Very thanks for your professional concern. We have fixed this typos.

Point 3: Page 2 3rd paragraph: Please check this phrase: "...suppressing the high sound pressure level source maybe not be able to improve the auditory feeling."

Response 3: Thank for you point out this obscure expression. Authors revise the original text to “The auditory feeling which can be quantified by those parameters maybe not be improved by suppressing the high sound pressure level source.”

Point 4: Pages 4 after Eq. 2.9: Please check this phrase: “More accurately reconstructed sound field scan be obtained if more microphones are used” More accurately reconstructed sound field be obtained if more microphones are used”

Response 4: Done as suggested.

Point 5: Please check this phrase: “For a the monopole sound source is located at point…”

Response 5: Done as suggested. The original text has been revised to “One monopole sound source is located at point….”

Point 6: Page 6, Last paragraph: “is not accurately enough” should be revise as “is not accurate enough”.

Response 6: Thank you for point out this typos. Done as suggested.

Point 7: Page 7 3.22 Errors: “reconstruction distant” should be revise as “reconstruction distance”.

Response 7: Thanks for your point out this typos. Done as suggested.

Point 8: Page 8, 1st paragraph: to with variations -> to variations

Response 8: Thanks for your point out this typos. Done as suggested.

Reviewer 2 Report

The authors investigate low-frequency sounds source localization for sound sources consisting of one and two monopoles. They use a reconstruction of the 3D volumetric sound field from the pressure measured on a sphere, as proposed by Williams et al. 2006 [11]. They use a rigid sphere with discrete microphones attached to its surface, for which the reconstruction formula was generalized by Jacobsen et al. 2011 [13]. Because the method only works accurately in the vicinity of the sphere it is called spherical near field acoustic holography (SNAH). The authors show the reconstruction errors of sound pressure, loudness and sharpness by comparison to the available analytical solution as a function of distance to the microphone sphere array and as a function of the truncation of the spherical harmonics. Furthermore, the distribution of those reconstructed quantities is visualized in angular space, suggesting that one and two monopole sound sources can be identified.

Here some issues which should be addressed:

1) Replace William et al. with the correct name Williams et al.

2) Since you do not propose or derive the equations in “2.1 Theoretical model for spherical acoustic holography with a rigid spherical array” the original works [11] and [13] should be cited in this section.

3) Please clean up the notation and text around 2.10. “W is a matrix of 372 representing the ear response”? The goal is to map the m discrete frequencies which you get from FFT of the time-dependent sound pressure (how is this dependent on the sampling rate and duration for instance?) to 372 and 24 loudness values a loudness and the Zwicker loudness respectively. This should be written more clearly.

4) The first sentence in “3.1. Errors of SNAH” shows some typos, for instance, the sound speed is 340 m/s^2. Please check and correct.

5) It should be possible to express the findings in a unit free manner. So the results should be dependent on ratios of wavelength to microphone array diameter to reconstruction distance. Could you maybe express some of the results in this scale-invariant form? At least please provide a comment regarding this.

6) Figures 5, 6, 8, 10, 11 are definitely of insufficient quality. The numbers on the axes are blurred and are much too small to be read. Some unexplained symbols/boxes are shown in every figure on the lower right side. I guess they have no meaning for the things which are aimed to be shown and thus should be removed.

7) “4. Experiment”. “The sound pressure data was measured by a Bruel&Kjaer4958 microphone, which has 10-20 kHz frequency range”. You should write 10 Hz – 20 kHz, otherwise, it would mean 10 kHz – 20 kHz.

Even when I think that the manuscript potentially can be published after repairing the issues which I mentioned above I don’t think that this work fits into Materials. Therefore, I strongly suggest a transfer to a journal that fits in scope and readership. From my perspective, MPDI Applied Sciences or Acoustics would be good choices.

Author Response

Dear reviewers:

We express our sincere appreciation for your careful reading and helpful comments. Those comments are of great significance for revising and improving our article. In the revise paper, we discuss two application of this work to characterization the sound absorption [24] or insulation [25] properties of noise control acoustic material, we think that this paper will fit with the scope of the journal under topic "Characterization techniques". We have addressed the points according to your advice and have given the answers to the comments below.

Point 1: Replace William et al. with the correct name Williams et al.

Response 1: Thanks for your suggestion. Done as suggested.

Point 2: Since you do not propose or derive the equations in “2.1 Theoretical model for spherical acoustic holography with a rigid spherical array” the original works [11] and [13] should be cited in this section.

Response 2: Thanks for your concern. We add Ref.11, and Ref. 13 in begin of section, and add Ref. 13 before Eq. 2.6.

Point 3: Please clean up the notation and text around 2.10. “W is a matrix of 372 representing the ear response”? The goal is to map the m discrete frequencies which you get from FFT of the time-dependent sound pressure (how is this dependent on the sampling rate and duration for instance?) to 372 and 24 loudness values a loudness and the Zwicker loudness respectively. This should be written more clearly.

Response 3: Thank you for point the miscellaneous. The sampling rate is 51.2 kHz, which is higher enough for audible sound, and the duration for instance in experiment is depended on the average time of FFT, which at least is 20s seconds.

Equation 2.10 is the definition of Moore Loudness model which has 372 frequency filter and listed in ANSI 3.4-2007.

The original text has been revised to “W is a matrix of 372 specific loudness model, each one is corresponded to each critical band, and audible frequency range is divided into those 372 critical bands.”

Point 4: The first sentence in “3.1. Errors of SNAH” shows some typos, for instance, the sound speed is 340 m/s^2. Please check and correct.

Response 4: Thanks for your concern. We revise the typos in this sentence. “air density is 1.3 kg/m3, sound speed is 340 m/s, and sound strength Qs is 3.6×10-5 Pa.m .”

Point 5: It should be possible to express the findings in a unit free manner. So the results should be dependent on ratios of wavelength to microphone array diameter to reconstruction distance. Could you maybe express some of the results in this scale-invariant form? At least please provide a comment regarding this.

Response 5: Thanks for your concern.  As the wavelength of air vary with the sound frequency, we need to account the wide band (at least band limited) sound to calculate sound quality objective parameters by one fix dimension microphone array, it is impossible to express the results in a unit free manner as the wavelength is not a fixed value.

Point 6: Figures 5, 6, 8, 10, 11 are definitely of insufficient quality. The numbers on the axes are blurred and are much too small to be read. Some unexplained symbols/boxes are shown in every figure on the lower right side. I guess they have no meaning for the things which are aimed to be shown and thus should be removed.

Response 6: Thanks for your concern. Now, Figures 5, 6, 8,10,11 have been redrawing. Those figures show the plane expansion of the spherical calculation surface, The axes in each figure is same, the horizontal axis is longitude: -π~π, the vertical axis is latitude: -π/2~π/2. So, we put the explanation in titles of Figure 5, 6, 8,10,11, 

“The results show in form of the plane expansion of the spherical calculation surface, the horizontal axis is longitude: -π~π, the vertical axis is latitude: -π/2~π/2”.

Point 7: “4. Experiment”. “The sound pressure data was measured by a Bruel&Kjaer 4958 microphone, which has 10-20 kHz frequency range”. You should write 10 Hz – 20 kHz, otherwise, it would mean 10 kHz – 20 kHz.

Response 7: Corrected as suggestion.

Reviewer 3 Report

The research shown in the paper is, a complete work. For that reason, is difficult to find points or parts revisable, independently of that in the next lines I show different commentaries.

1 Introduction

First section of article is a good review of estate of art of research area. As the only comment, authors repeat some references on text, as example reference 15 2 time on the same paragraph. In my opinion is better to text only referee one time for reference but it is possible this consideration be difference for editors

In all of that section the authors set out in a very clear and summary way the objective of the work and it has not corrections for to be publish

2 Methods

On Methods section, authors have shown step by step the mathematical development used on work, being it very clear to understand. That section is correct to be published

3 Simulation

The chapter of simulation is very complete and it show a lot of details. At this point it has 2 commentaries:

On charts, yellow lines are very difficult to see. It is recommended change that colour to improve the visibility of different lines.

Simulations and explanations of these point are clear and very visuals on paper, and the differences of simulations are easy to identify. That part of paper has a big quality on my opinion as a reviewer.

4 Experiment and 5 Conclusions

First point is correct to publish, it shows the experimental process to validate the simulation with a good resolution on figures and different cases.

And on the other hand, on conclusion section authors present a short summary of research. On last part authors write about “limitations” of methods and it would be interesting to to explain “future lines or steps” of investigation to continue the research shown of this pape.

In general, the article is right to publish with a minor review, even without take in account the commentaries because its are not very important, due to the good procedure of work.

Author Response

Dear Reviewer:

We express our sincere appreciation for your careful reading and helpful comments. Those comments are of great significance for revising and improving our article. We have addressed the points according to your advice and have given the answers to the comments below.

Point 1: Put reference 15 2 time on the same paragraph

Response: Thanks for your concern. Second Reference 15 cited has been deleted

Point 2: Simulation Section: On charts, yellow lines are very difficult to see. It is recommended change that color to improve the visibility of different lines.

Response 2: Thank you for point out the in charts, the yellow lines in Figure 2, 3 and 4, have been revised to dark blue line.

Point 3: Conclusion: explain “future lines or steps” of investigation to continue the research shown of this paper.

Response 3: Thank you for point out the in charts, we put the description of the on-going work in the end of paper, which is” ……..in more complex acoustic environments, both of them are further being studied in the laboratory”

Round 2

Reviewer 2 Report

The manuscript was revised satisfactory. The collective perception of the other reviewers and the editor should overrule my outlier perception regarding the journal choice. The work is of sufficient interest and quality. Therefore, it deserves to be published.